# Surgical Correction of a Sinus Venosus Atrial Septal Defect with Partial Anomalous Pulmonary Venous Connections Using Cardiac Computed Tomography Imaging and a 3D-Printed Model

**DOI:** 10.3390/ani14071094

**Published:** 2024-04-03

**Authors:** Kyung-Min Kim, Chang-Hwan Moon, Won-Jong Lee, Woo-Jin Kim, Mihyung Kim, Jaemin Jeong, Hae-Beom Lee, Seong-Mok Jeong, Ho-Jung Choi, Tae Sung Hwang, Hee Chun Lee, Jae Hyeon Yu, Aryung Nam, Dae-Hyun Kim

**Affiliations:** 1Department of Veterinary Surgery, College of Veterinary Medicine, Chungnam National University, 99 Daehak-ro, Yuseong-gu, Daejeon 34134, Republic of Korea; kgmtong8886@daum.net (K.-M.K.); moonch0208@gmail.com (C.-H.M.); wjl03ssaaa@naver.com (W.-J.L.); loneless1175@naver.com (W.-J.K.); klmie800@cnu.ac.kr (J.J.); seatiger76@cnu.ac.kr (H.-B.L.); jsmok@cnu.ac.kr (S.-M.J.); 2Chungnam National University Sejong Hospital, 20 Bodeum 7-ro, Sejong-si 30099, Republic of Korea; 33sebomi@naver.com; 3Department of Veterinary Medical Imaging, College of Veterinary Medicine, Chungnam National University, 99 Daehak-ro, Yuseong-gu, Daejeon 34134, Republic of Korea; hjchoi@cnu.ac.kr; 4Institute of Animal Medicine, College of Veterinary Medicine, Gyeongsang National University, 501 Jinju-daero, Jinju-si 52828, Republic of Korea; hwangts@gnu.ac.kr (T.S.H.); lhc@gnu.ac.kr (H.C.L.); 5Department of Cardiothoracic Surgery, Chungnam National University Hospital, College of Medicine, Chungnam National University, 282 Munhwa-ro, Jung-gu, Daejeon 35015, Republic of Korea; jahyu@cnu.ac.kr; 6Department of Veterinary Internal Medicine, College of Veterinary Medicine, Konkuk University, 120 Neungdong-ro, Gwangjin-gu, Seoul 05029, Republic of Korea

**Keywords:** sinus venosus atrial septal defect, partial anomalous pulmonary venous connection, 3D-printed cardiac model, cardiopulmonary bypass, congenital heart disease, dog

## Abstract

**Simple Summary:**

This report describes the case of a sinus venosus atrial septal defect, concurrent with partial anomalous pulmonary venous connections, which is a rare congenital heart disease in dogs. In patients with hemodynamically significant changes and clinical signs, treatment of the defect is essential. There are various treatment options for an atrial septal defect, including surgical correction under cardiopulmonary bypass or inflow occlusion, and interventional closure with occluders. In this case, given the significant size of the defect and the presence of concomitant vascular anomalies, performing closure with a pericardial patch under a cardiopulmonary bypass is considered as the most suitable approach. Additionally, employing a cardiac model from computed tomography imaging facilitated a preoperative simulation, enhancing the precision and efficiency of the surgical procedure, both in terms of accuracy and time. To the best of our knowledge, this is the first reported case in dogs with a sinus venosus atrial septal defect, accompanied by partial anomalous pulmonary venosus connections, where a computed tomography-based three-dimensional cardiac model was created to precisely identify the defect’s location and associated cardiovascular anomalies, leading to a successful surgical intervention under a cardiopulmonary bypass.

**Abstract:**

Sinus venosus atrial septal defects (SVASDs), concurrent with partial anomalous pulmonary venous connections (PAPVCs), are a rare congenital heart disease in dogs. Surgical correction is essential when clinical signs or significant hemodynamic changes are present. We aimed to report on the successful surgical correction of an SVASD with PAPVCs, using a computed tomography (CT)-based customized 3D cardiac model. A 10-month-old male poodle was referred for corrective surgery for an ASD. Echocardiography confirmed a hemodynamically significant left-to-right shunting flow through an interatrial septal defect and severe right-sided heart volume overload. For a comprehensive diagnosis, a CT scan was performed, which confirmed an SVASD with PAPVCs. A customized 3D cardiac model was used for preoperative decision-making and surgical rehearsal. The defect was repaired using an autologous pericardial patch under a cardiopulmonary bypass (CPB). Temporary pacing was applied for sinus bradycardia and third-degree atrioventricular block. The patient recovered from the anesthesia without further complications. The pacemaker was removed during hospitalization and the patient was discharged without complications 2 weeks post-surgery. At the three-month follow-up, there was no shunting flow in the interatrial septum and the right-sided volume overload had been resolved. The cardiac medications were discontinued, and there were no complications. This report indicates the validity of surgical correction under CPB for an SVASD with PAPVCs, and the advantages of utilizing a CT-based 3D cardiac model for preoperative planning to increase the surgical success rate.

## 1. Introduction

Atrial septal defects (ASDs) are a rare congenital cardiovascular defect in dogs that allow for interatrial blood flow [1]. There are four major types of ASD, categorized by the location of the defect in the interatrial septum: ostium secundum, the most common type of ASD is located in the middle of the septum and involves the fossa ovalis; primum ASD, which is located in the most apical portion of the septum and may coexist with ventricular septal defects; sinus venosus ASD, a rare type of ASD, which is in the uppermost portion of the septum, close to the junction of the cranial or caudal vena cava; and coronary sinus ASD, which is located between the wall of the coronary sinus and left atrium [2,3].

In dogs with small defects, hemodynamic changes may be insignificant, and the corresponding clinical signs may not exist. However, with large defects, a significant left-to-right shunt results in volume overload in the right ventricle (RV) and subsequent right ventricular dilatation, which increases pulmonary blood circulation [4]. These changes induce pulmonary hypertension or right heart failure, with associated respiratory compromise and exercise intolerance. Surgical correction is indicated in cases with hemodynamically significant changes, clinical signs, and associated partial anomalous pulmonary venous connections (PAPVCs) [4,5,6].

In humans, 90% of sinus venosus ASDs (SVASDs) are associated with concurrent PAPVCs [7], a rare congenital vascular anomaly in which one or more pulmonary veins drain into the right atrium (RA), RA-vena cava junction, or cranial or caudal vena cava (CrVC or CdVC). In patients with PAPVCs, echocardiography may be insufficient for a definitive diagnosis of pulmonary vein anomalies, potentially leading to misdiagnosis and inappropriate treatment [8]. Therefore, comprehensive imaging examinations, such as computed tomography (CT) scans, should be performed, particularly in patients with complex congenital cardiovascular anomalies. These examinations identify intracardiac structures and extracardiac pulmonary and systemic arterial and venous cardiovascular structures [9].

Electrocardiogram (ECG)-gated cardiac CT imaging, in addition to conventional CT imaging, not only reduces motion artifacts, but also enables the precise evaluation of cardiac structures, which is especially beneficial for congenital anomalies [10]. This data offers supplementary insights into surgical anatomy, enhancing the understanding of physical cardiac structures for surgical interventions [9]. However, the interpretation of image data on virtual two-dimensional (2D) monitors may be insufficient to comprehend the intricate geometry of the 3D heart. CT-based 3D-printed models offer a more accurate physical visualization of spatial intracardiac anatomy and extracardiac vascular structures. These personalized physical reality models enable surgeons to rehearse surgical procedures and determine optimal approaches for patients and the specific locations of the anomalies. This facilitates preoperative decision-making and surgical planning, enhancing overall surgical success [11,12].

In cases of SVASDs concurrent with PAPVCs, it is important to note that ASD corrective treatment alone may not fully resolve the clinical signs [8]. Therefore, a comprehensive diagnosis and precise surgical approach is necessary for each confirmed anomaly to improve the clinical symptoms and minimize postoperative complications, contributing to a positive prognosis. We, therefore, aimed to report on the successful surgical correction of an SVASD concurrent with PAPVCs, using an autologous pericardial patch. Additionally, we highlight the utility of CT-based 3D-printed cardiac models for congenital cardiovascular anomalies, in conjunction with traditional echocardiography for diagnosis.

## 2. Case

A 10-month-old male toy poodle was referred to the veterinary medical teaching hospital at Chungnam National University for surgical correction of an ASD. The patient presented with dyspnea, a cough, exercise intolerance, and lethargy. The initial treatment at the primary animal hospital involved pimobendan (0.25 mg/kg, orally, twice a day), sildenafil (1.5 mg/kg, orally, twice a day), and furosemide (0.5 mg/kg, orally, twice a day) for the ASD and suspected pulmonary hypertension, based on echocardiography.

During the physical examination, a systolic ejection murmur of grade 4/6 was auscultated at the heart base. Blood analysis showed no specific abnormalities and the cardiac dysfunction biomarkers, including Troponin I (<0.2 ng/mL) and N-terminal pro-B-type natriuretic peptide (508 qmol/L; reference range 0–900), were within the reference range.

Regarding thoracic radiography, the overall cardiomegaly, particularly on the right side, with a vertebral heart score of 10.5, and pulmonary vein dilatation, were confirmed. Two-dimensional echocardiography revealed severe enlargement of the RV, with a calculated RV:LV (left ventricle) ratio of 3:2 (Figure 1A). Mild tricuspid regurgitation, pulmonary trunk dilatation, and interventricular septal flattening were observed, and mild pulmonary hypertension was suspected [13]. The right parasternal long-axis four-chamber view identified a 3.8 mm defect at the upper edge of the atrial septum (Figure 1B) and additional shunt flow, which was observed as linear echogenic duct-like structures opening into the RA. Color Doppler imaging revealed left-to-right shunting of the blood flow through the defect at a flow velocity of 1.6 m/s during ventricular systole (atrial diastole) (Figure 1C). The calculated pulmonary to systemic flow ratio (Qp:Qs), using the velocity time integral and cross-sectional area, was 3:1, which was a hemodynamically significant change and supported the presence of a large left-to-right shunting flow [4,14,15].

These echocardiographic findings suggested that addressing the ASD alone would be insufficient, given the severe RV volume overload, the additional blood flow opening into the RA, and the significant hemodynamic changes. In addition, to evaluate the shunting flow into the RA, a saline bubble test was conducted. Following the administration of agitated saline via the cephalic vein, strong bubbles were observed within the right and left atria, along with the duct-like structure. This suggests communication between the left atrium and cranial vena cava, with the duct-like structure likely associated with the cranial vena cava. Assuming a sinus venosus ASD, the observed shunt flow was suspected as PAPVCs. 

The more accurate evaluation of the defect and blood flow were assessed using a 160-multislice CT scanner (Aquilion Lightning 160^®^, Canon Medical Systems, Otawara, Japan). The CT scans were acquired using the non-gated ECG CT technique, and the scanning parameters were adjusted, as follows, to minimize motion blurring artifacts from the heart: 120 kV, 200 mA, gantry speed of 0.75 s/rotation, slice collimation of 0.5 × 80 mm, 0.5 mm slice thickness, and a pitch factor of 1.388. The CT scans confirmed the presence of an ASD in the upper portion of the atrial septum, near the cranial vena cava. Additionally, the right cranial and middle pulmonary veins were inserted into the RA. Considering all the echocardiography and CT findings, the diagnosis of an SVASD with PAPVCs was established (Figure 1D,E).

To enhance the visualization of intra- and extra-cardiovascular anomalies, we employed 3D printing technology to create a cardiac model. The CT data were transferred to a 3D printing company (Anymedi Inc., Seoul, Republic of Korea) and printed with different colors, using flexible, rubberlike, and translucent materials (Objet500 Connex3; Stratasys, Minnesota, MN, USA) [11]. This 3D-printed cardiac model provided a comprehensive spatial representation of the structures associated with the right cranial pulmonary vein (RCrPV) and right middle pulmonary vein (RMPV), both of which drained into the RA (Figure 2A). Additionally, the model allowed the visualization of the location of the interatrial defect, through a window in the right auricle (Figure 2B,C). Given the diagnosis of an SVASD with PAPVCs, it was determined that surgical correction under a cardiopulmonary bypass (CPB) was the most appropriate option for the patient [4,7].

The patient was premedicated with fentanyl (Myungmoon fentanyl citrate injection 1 g; Chong Kun Dang Pharmaceutical Corp., Seoul, Republic of Korea; 5 µg/kg IV) and midazolam (Bukwang midazolam injection; Bukwang Pharmaceutical Co., Ltd., Seoul, Republic of Korea; 0.25 mg/kg IV). The patient was slowly administered propofol (anepol injection; Hana Pharmaceutical Co., Ltd., Seoul, Republic of Korea; 3 mg/kg IV, slowly), then intubated with an endotracheal tube. Anesthesia was maintained with isoflurane (Ifran Liquid for Inhalation; Hana Pharmaceutical Co., Ltd., Seoul, Republic of Korea) and, during CPB, total intravenous anesthesia of propofol (0.2–0.5 mg/kg/min IV) was administered without an inhalant agent. During the CPB weaning stage, anesthesia was gradually maintained and, simultaneously, returned to isoflurane. Bupivacaine (Myungmoon bupivacaine HCl 0.5%; Myungmoon Pharm. Co., Ltd., Seoul, Republic of Korea; 1 mg/kg) was infiltrated into the right 5–6th intercostal space for intercostal block, and fentanyl (5 µg/kg/hr) was infused throughout the surgical procedure and for postoperative pain management. Anesthesia was monitored continuously, including the heart rate, respiratory rate (RR), arterial oxygen saturation (SpO_2_), end-tidal CO_2_ (EtCO_2_), non-invasive blood pressure (NIBP), invasive blood pressure, rectal and esophageal temperatures, and isoflurane concentrations, using an anesthesia monitoring machine.

The patient was placed in the left recumbent position for the surgical incision. Initially, the femoral triangle approach was performed for the arterial line (A-line) placement. The femoral artery was catheterized (22-G indwelling cannula with three-way stopcock) to monitor the continuous systolic, diastolic, and mean arterial blood pressure measurements. Additionally, femoral artery catheterization provided arterial blood samples for blood gas, activated clotting time (ACT), and complete blood count analysis.

Following A-line catheter placement, a right thoracotomy was performed at the fifth intercostal space. A partial pericardiectomy was performed for defect closure, and the excised pericardium was immersed in a 0.625% glutaraldehyde (GA) solution for 15 min, followed by three rinses in normal saline. The prepared pericardium was stored for subsequent applications.

For the CPB preparation, the ascending aorta and the cranial and caudal vena cava were isolated for arterial and venous cannulation. The aortic root was isolated for subsequent cardioplegic injections. Initially, 250 U/kg heparin was administered to achieve an ACT of >350 s, followed by an additional 100 U/kg heparin. Once the ACT was confirmed to be >300 s, a 12-Fr arterial cannula was inserted into the ascending aorta. The 12-Fr and 14-Fr venous cannulas were inserted, and each was secured with a purse-string suture, using a 6–0 polypropylene suture (Prolene 6–0; Ethicon, Johnson & Johnson Company, Somerville, NJ, USA), placed with a tourniquet for fixation.

The prepared CPB circuit was roller-type machine equipped with an oxygenator (Terumo Capiox^®^ FX05; Terumo Co., Tokyo, Japan). The tubing sizes were 3/16 for the arterial line and 1/4 for the venous line. The CPB circuit was primed with a solution of 20% albumin (100 mL), 20% mannitol (5 mL/kg), 8.4% sodium bicarbonate (1 mL/kg), heparin (500 U), and cefazolin (22 mg/kg), packed red blood cells to maintain a hematocrit (Hct) level of 20–25%, and a Plasma-Lyte solution was used as a volume expander. CPB was partially initiated when the measured ACT exceeded 350 s, and anesthesia maintenance was switched from isoflurane inhalation to continuous intravenous infusion. A 4-Fr (18-G) root cannula was inserted into the aortic root with a purse-string suture, using a 6–0 polypropylene suture. The aorta was occluded using an aortic cross-clamp, and total CPB was initiated immediately. Cold cardioplegia (cardioplegic solution 1; JW pharmaceutical Co., Gyeonggi-do, Republic of Korea) mixed with blood from the CPB circuit, in a 1:2 ratio, was administered rapidly to induce cardiac arrest, followed by the subsequent administration of ice every 20 min.

For the intracardiac approach, a right atriotomy was performed at the right atrial appendage (RAA). An approach to the defect was established, based on preoperative visualization and surgical plans aided by the 3D-printed cardiac model. Subsequently, the SVASD was confirmed through the RAA window (Figure 3A). The openings of the RCrPV and RMPV were craniodorsal to the ASD and both veins were emptied into the RA (Figure 3B). The closure of the ASD and PAPVCs were performed concurrently using an autologous pericardium patch that had been fixed with GA (Figure 3C). The patch was sutured to the atrial septal wall using a 6–0 polypropylene suture (Figure 3D), creating a space to secure distance from the anomalous pulmonary vein and allow blood flow into the left atrium.

Before completing ASD closure, the patient’s lungs were manually inflated to eliminate any air in the LA. The RA was then closed with a 6–0 polypropylene suture, and the patient was gradually rewarmed. The ACC time was 55 min, and the patient was de-clamped when the body temperature reached 34 °C. As the body temperature gradually increased, the CPB flow was slowly reduced to restore cardiac function. A total of 3 mg/kg of protamine (Hanlim protamine sulfate injection; Hanlim Pharmaceutical Co., Ltd., Seoul, Republic of Korea) was administered intravenously for 30 min to antagonize the effects of the heparin. The cannulas in the CrVC, CdVC, and the arterial cannula were removed and ligated individually, and the femoral artery was ligated after the intravenous catheter was removed. 

While gradual weaning from the CPB and restarting the heart progressed, fibrillation was observed on the echocardiography (ECG). Through the application of a defibrillator, and direct cardiac massage, warm blood shot through the aortic root cannula along the coronary artery, and through the additional administration of emergency medications, resuscitation was accomplished. However, continuous sinus bradycardia, and intermittent third-degree AV block persisted after resuscitation. Additionally, despite the application of a defibrillator to restore the ECG, arrhythmia recurred. Owing to persistent arrhythmias, each temporary pacing wire was placed on the epicardium of the RV and right thoracic wall (VVI mode, unipolar epicardial pacing). After chest tube insertion, the thoracotomy incision was closed using standard techniques. Intraoperative transesophageal echocardiography confirmed the placement of the interatrial patch and assessed the cardiac contractility before complete closure of the chest incision.

The patient recovered smoothly from anesthesia, with continuous monitoring of the ECG parameters, including the heart rate, RR, SpO_2_, EtCO_2_, NIBP, and rectal temperature, throughout the recovery period. The ECG waveforms were influenced by pacing, but gradually recovered. The pacing rates and degrees were meticulously controlled by the monitoring personnel. No cardiac murmurs were auscultated, and the respiratory auscultation was normal immediately after surgery.

Postoperative echocardiography confirmed the closure of the defect with no residual interatrial blood shunt. During hospitalization, the pacing rates delivered by the temporary pacemaker were controlled and gradually tapered. Five days post-surgery, pacing was discontinued and, on the seventh day, the pacing wires were smoothly removed by gently pulling them without additional sedation. The patient was discharged 2 weeks postoperatively. The postoperative medications were pimobendan (Vetmedin; Boehringer Ingelheim, Germany; 0.25 mg/kg, orally), twice daily, and rivaroxaban (Xarelto; Bayer Yakuhin, Ltd., Osaka, Japan; 0.5 mg/kg, orally), once daily, for 3 months post-surgery.

At the 3-month follow-up, thoracic radiography revealed a vertebral heart size (VHS) of 10.3, with a significantly decreased cardiac proportion within the thoracic cavity compared with the preoperative assessment (Figure 4A,B). Echocardiography indicated the absence of residual interatrial flow and the adjusted patch remained appropriately positioned. The Qp:Qs ratio was 0.91:1 and the RV volume overload returned to the normal range (Figure 4C,D). Therefore, the cardiac oral medications were discontinued.

At the 12-month follow-up, the patient continued to exhibit good vitality and appetite. Thoracic radiography showed a similar VHS, with a further reduction in the cardiac proportion, when compared to the previous assessment. Echocardiography confirmed the absence of any residual shunt between the atria. The ratio of the left heart to the right heart size was 2:1, indicating a significant decrease in the right heart volume overload compared to the preoperative assessment. The other echocardiographic parameters were similar to those at the previous follow-up.

## 3. Discussion

This report describes the successful surgical repair of an SVASD concurrent with PAPVCs in a dog, using direct open-heart surgical techniques under a CPB. Immediately following surgery, the interatrial septal blood shunt disappeared, and the dog showed no clinical signs of cardiovascular complications, even after the discontinuation of all cardiovascular medications.

In veterinary medicine, an ASD is considered a rare congenital heart disease (CHD), with limited information regarding surgical correction. In the case of an SVASD, corrective surgery necessitates the use of CPB for arrested open-heart surgery. Given the anatomical and physiological complexities of this specific defect, surgical interventions may need to be tailored according to comprehensive treatment plans and procedural decision-making [7]. Although interventional treatments are available using occluders for interatrial defect correction, these devices can only be applied in a secundum-type ASD without concurrent PAPVCs or other vascular anomalies [5,8,16]. Particularly in the case of an SVASD, in which 85–90% of patients have concurrent PAPVCs, a more comprehensive diagnostic examination is essential [7,17,18]. During surgery, the routine assessment of the pulmonary vein drainage site is crucial to exclude PAPVCs, which could be misdiagnosed or overlooked during diagnosis [7,19].

In this case, even when considering left-to-right blood flow through the ASD, the patient exhibited excessive right heart distension (LV:RV = 2:3). Echocardiography confirmed abnormal blood flow into the RA. Given the challenging nature of diagnosing anomalous pulmonary venous drainage and congenital cardiovascular anomalies, along with the potential for misdiagnosis, CT or magnetic resonance imaging has been increasingly adopted [12]. CT enables the confirmation of intracardiac defects, extracardiac vascular structures, and their connections. Utilizing CT dataset 3D reconstruction methods, 3D volume-rendered images have become valuable tools for diagnosing CHDs [9,20]. However, despite the advantages of these methods in enabling observers to manipulate a patient’s heart from various angles, they are still limited by the reliance on interpretation from a 2D monitor [21], hindering a comprehensive understanding of intracardiac structures [22]. To overcome this limitation, significant developments have been made in the creation of customized 3D heart models generated from a patient’s CT imaging data [23,24]. Through personalized 3D heart models, surgeons enhance their spatial understanding of cardiac structures and make more precise determinations regarding the location of the right atrial incision and the anatomical structures requiring correction [12,21]. These advantages are particularly noteworthy in veterinary medicine because of the smaller size of the heart compared to a human heart, which hinders optimal visibility during surgery. Creating additional incisions in the heart for a wider field of visualization and defect confirmation is limited due to the risk of damage to the myocardium and surrounding nodes [25]. Therefore, pre-visualization of the defect and the site of the pulmonary vein opening is crucial. The use of patient-specific 3D heart models significantly reduces the decision-making time during surgery and allows for a shorter surgical duration under CPB [21]. This contributes to an increased success rate and more refined surgical outcomes.

The surgical repair of a CHD is associated with the highest likelihood of postoperative arrhythmias, among the expected complications. Atrial fibrillation/flutter, atrioventricular block, and sinus bradycardia are common; if these manifest during surgery, pacemakers are used to correct these arrhythmias [26]. Furthermore, other arrhythmias, such as bradyarrhythmia, sinus node dysfunction, and supraventricular tachycardia, can occur and may be managed through assisted pacemaker control [27]. Epicardial or transvenous temporary cardiac pacing wires can be adjusted during surgery and gradually tapered until removal during postoperative hospitalization. If these arrhythmias persist for more than 14 days, replacement with a permanent pacemaker is indicated [27]. Therefore, in elective surgery to repair CHDs, including ASDs, the surgical team should be prepared for perioperative cardiac arrhythmias.

## 4. Conclusions

Surgical correction is imperative for SVASDs with concurrent PAPVCs in dogs, when hemodynamically significant changes occur. Unlike other types of ASD, cases with concurrent pulmonary vein anomalies present greater challenges for surgical correction, emphasizing the significance of a comprehensive diagnosis and a precise understanding of the anatomical structures. This report highlights a successful surgical correction using a CT-based, customized, 3D-printed heart model for rehearsal surgery, resulting in favorable surgical outcomes.

## Figures and Tables

**Figure 1 animals-14-01094-f001:**
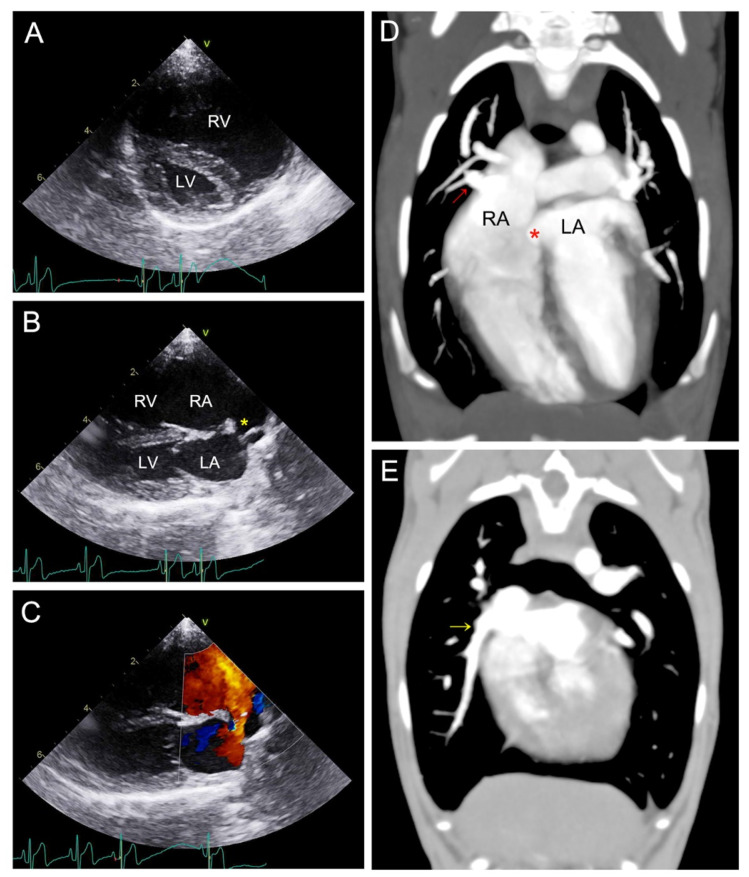
Preoperative echocardiographic images and computed tomography angiography. Right parasternal short-axis view (**A**) and long-axis (**B**,**C**) views of the heart. The LV:RV ratio is 2:3 (**A**), which means the right atrium and ventricle are significantly enlarged (**B**). The yellow star indicates an interatrial septal defect (**B**) and, in color Doppler mode, a left-to-right shunting flow via the defect is confirmed (**C**). The long-axis view of computed tomography angiography indicates partial anomalous pulmonary venous connections. The right cranial pulmonary vein (red arrow), which drains into the right atrium and interatrial septal defect (red star) were confirmed (**D**). Also, an additional right middle pulmonary vein draining into the RA (yellow arrow) was confirmed (**E**). RA, right atrium; LA, left atrium; LV, left ventricle; RV, right ventricle.

**Figure 2 animals-14-01094-f002:**
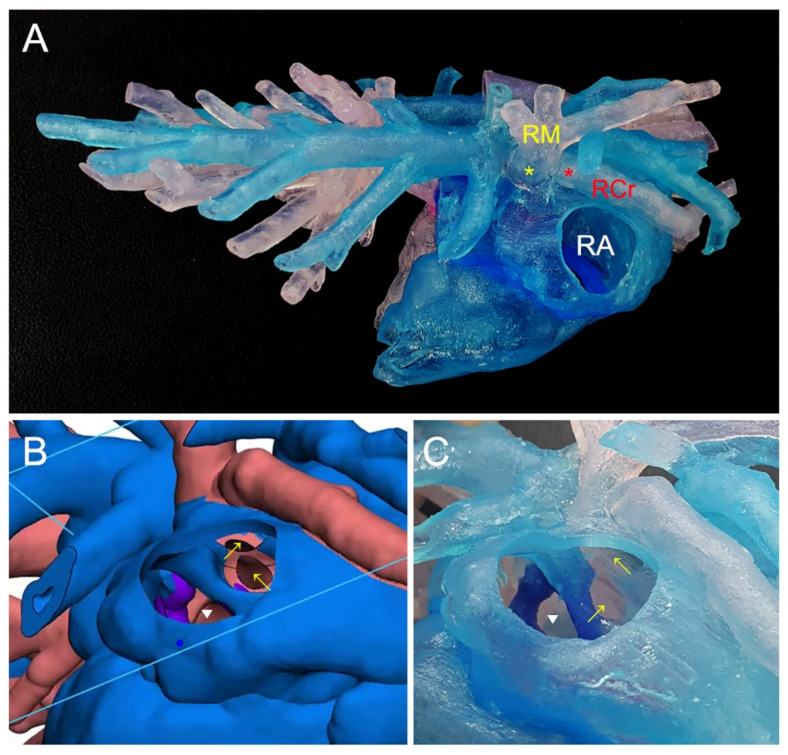
CT image-based, patient customized, 3D-printed cardiac model. The 3D-printed model demonstrates that the right middle pulmonary vein and right cranial pulmonary vein drain into the RA (**A**). Using virtual production software, the PAPVCs (yellow arrows) and ASD (white arrowhead) were confirmed through the right atrium window (**B**). Opening the window at the right atrium in the real 3D model, the right cranial and middle pulmonary vein openings (yellow arrows) were confirmed on the craniodorsal side of the ASD (**C**). RM, right middle pulmonary vein; RCr right cranial pulmonary vein; RA, right atrium.

**Figure 3 animals-14-01094-f003:**
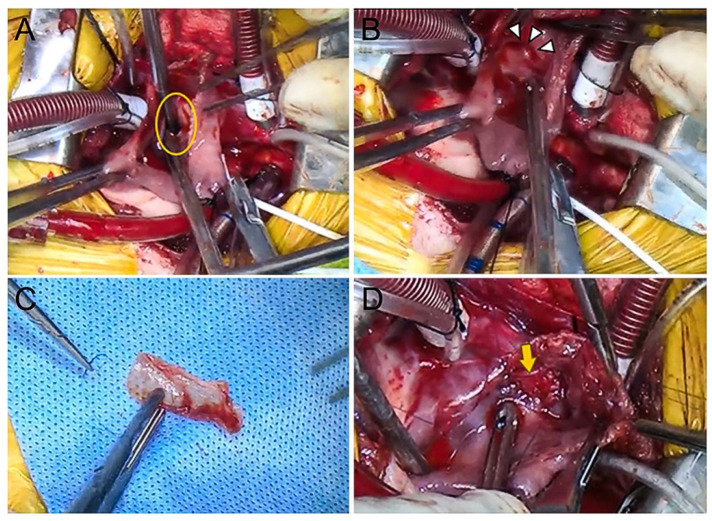
Intraoperative images. The SVASD (yellow circle) is confirmed through the open RA window (**A**). The anomalous pulmonary vein openings (white arrowheads) are confirmed on the craniodorsal side of the ASD (**B**). A trimmed glutaraldehyde-fixed autologous pericardial patch is prepared for the defect closure (**C**). Using a 6–0 polypropylene suture, the pericardial patch is adjusted at the defect. Almost completed closure of the defect with the patch (yellow arrow) (**D**).

**Figure 4 animals-14-01094-f004:**
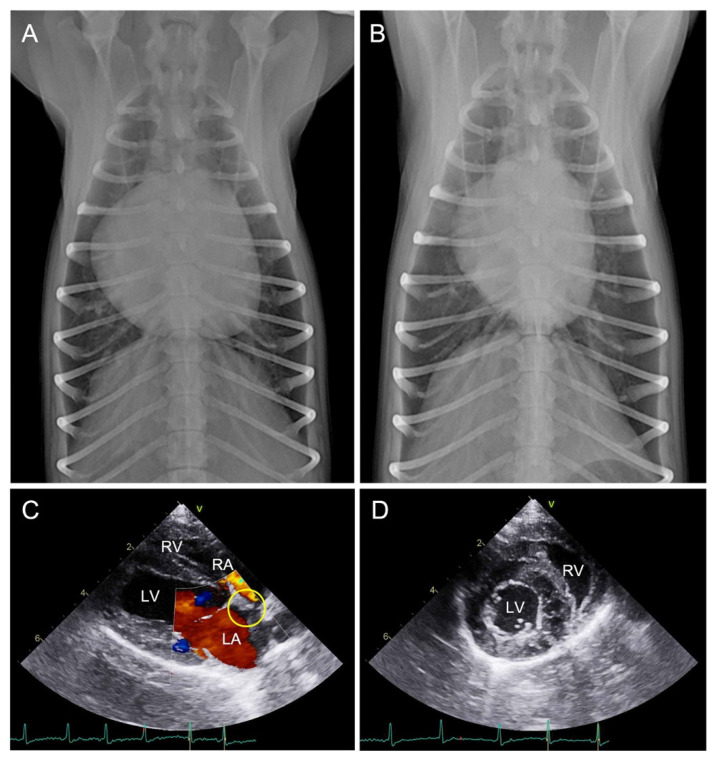
Radiographic images before (**A**) and 104 days after surgery (**B**). A significant decrease in the heart proportion 3 months after surgery (67%) is confirmed within the thoracic cavity compared to the preoperative (73%) results. Echocardiographic images 3 months after surgery (**C**,**D**). Right parasternal long-axis four-chamber view (**C**) and short-axis view (**D**) of the heart. In the color Doppler, no residual interatrial shunting flow through the patch location (yellow circle) is confirmed (**C**). The right heart size is reduced, and the LV:RV ratio return to normal (**D**). LV, left ventricle; LA, left atrium; RV, right ventricle; RA, right atrium.

## Data Availability

The original contributions presented in this study are included in the article, further inquiries can be directed to the corresponding author.

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
