# Peer review of "Surgical Correction of a Sinus Venosus Atrial Septal Defect with Partial Anomalous Pulmonary Venous Connections Using Cardiac Computed Tomography Imaging and a 3D-Printed Model"

_animals, 2024, doi:10.3390/ani14071094_

Round 1

Reviewer 1 Report

Comments and Suggestions for Authors

I believe this case report is indeed valuable, but there are several important comments to be made.

1. The imaging conditions for the CT scan are not clearly indicated. Is this cardiac CT a cardiac-gated CT? If not, the size of the obtained CT images would likely be ambiguous.

2. It is not clear how the actual creation of a three-dimensional model using a 3D printer, rather than utilizing 3D volume rendering, provides benefits.

3.Why do you think surgical correction under cardiopulmonary bypass (CPB) was the sole and most appropriate option for the treatment of Sinus Venosus Atrial Septal Defect with Partial Anomalous Pulmonary Venous Connection? Does the CT contribute to this decision?

Author Response

We would like to express our gratitude to the editor and reviewers for taking the time to carefully read and to consider the report on our present study. We appreciate the opportunity to revise the manuscript by addressing the reviewers’ comments based on their constructive guidance to help you and the reviewers in your final decision. Below, we have provided detailed and concise point-by-point responses to each of the reviewer concerns in conjunction with the revised manuscript. If I answered the questions you asked without making changes to the manuscript or in response to the questions, the words or sentences in blue colored words or sentences indicate my responses. Our responses in this document and the corresponding changes in the manuscript are written in red.

I have attached the responses to the comments you provided in a Word file.

Reviewer 2 Report

Comments and Suggestions for Authors

Thank you for the opportunity to review this report.  The authors are commended for successful management of a challenging congenital defect.  While repair of sinus venous ASD is not in itself novel, the use of a 3D printed model for surgical planning does appear to be novel and worthy of publication.  In general, the report reads fairly well, but some detail could be added to help strengthen the report.  Since this is a single case report that does not require in-depth analysis or statistics, I would recommend a minor revision prior to publication.  I have attached a highlighted and commented PDF with specific comments, questions, and suggestions to consider for a revised version.   

Comments on the Quality of English Language

The quality of the English in this report is good.

Author Response

We would like to express our gratitude to the editor and reviewers for taking the time to carefully read and to consider the report on our present study. We appreciate the opportunity to revise the manuscript by addressing the reviewers’ comments based on their constructive guidance to help you and the reviewers in your final decision. Below, we have provided detailed and concise point-by-point responses to each of the reviewer concerns in conjunction with the revised manuscript. If I answered the questions you asked without making changes to the manuscript or in response to the questions, the words or sentences in blue colored words or sentences indicate my responses. Our responses in this document and the corresponding changes in the manuscript are written in red.

I have attached the responses to the comments you provided in a Word file.

Sincerely,
Kyung-Min Kim

Reviewer 3 Report

Comments and Suggestions for Authors

Reviewers comments

Surgical Correction of a Sinus Venosus Atrial Septal Defect  with Partial Anomalous Pulmonary Venous Connection using Cardiac Computed Tomography Imaging and a 3D-Printed Model

In general

This is a really interesting case report of a seldom seen ASD type an the concurrent vascular anomalies. It is not the first case described with this combination, but with the use of the 3-D model it is defenitely new an the complications seen in this patient are important to describe because most of our information is from human medicine.

It would be great to describe the primarilay echo mor into detail espacially the other views than only long axis, if available also 3-D echo datas.

If these informations are added it should be a case to be published

Simple summary

fine

Abstract

Fine

Introduction

Line 65: Please add the 4. ASD form: coronary sinus type

Case

Line 115: Could you please add the values of the biomarkers (NT-pro BNP would be interresting to see the level because in this case there is a volume overload of the right heart and a suspected pulmonary hypertension)

Line 119: left ventricle ist not enlarged it is smaller than normal, right atrium is enlarged, please correct this

Line 121: Please describe in which phase a flattening was seen

Line 124: Please specifiy also here at what time point peak velocity was seen

Line 126: Qp/Qs is high, what was the velocity over the pulmonic valve, because this a very nice easy parameter to see if there is a volume problem.

Line 128: Was is possible to see the defect in the short axis view and describe as a defect to the cranial vena cava? In a three dimensional way this defect could be large enough to make this kind of volume overload, please discuss this later.

Line 134: The question is would it be possible to find thes PAPVC in the echo. Please correlate these data with the human medicine.

Line 241: How was the pacing wire secured at the peicardium, that it was possible to take it away after 5 Days?

Line 295: The information about prevalence of PAPVC is human? Please the reference. Are there any informations for dogs? Please discuss this fact

Author Response

(The authors gave the same response as above.)

Round 2

Reviewer 1 Report

Comments and Suggestions for Authors

I think the points raised last time have been largely addressed. Given that this case is a rare case of treatment for a unique cardiac anomaly, I believe it holds valuable significance as a case report.